# Modulation of host lipid metabolism by virus infection leads to exoskeleton damage in shrimp

Xin-Xin Wang[1,2,3], Ming-Jie Ding[1,2,3], Jie Gao[1,2,3], Ling Zhao[4], Rong Cao[4], Xian-Wei Wang [1,2,3]*

1 Shandong Provincial Key Laboratory of Animal Cells and Developmental Biology, School of Life Sciences, Shandong University, Qingdao, China, 2 Laboratory for Marine Biology and Biotechnology, Qingdao Marine Science and Technology Center, Qingdao, China, 3 State Key Laboratory of Microbial Technology, Shandong University, Qingdao, China, 4 Department of Food Engineering and Nutrition, Yellow Sea Fisheries Research Institute, Chinese Academy of Fishery Sciences, Qingdao, China

* wangxianwei@sdu.edu.cn

**Data Availability Statement:** All relevant data are within the paper and its Supporting Information files.

**Funding:** This work was supported financially by the Natural Science Foundation of China (32173008, 32373159) to XWW. The funders have

## Abstract

The arthropod exoskeleton provides protection and support and is vital for survival and adaption. The integrity and mechanical properties of the exoskeleton are often impaired after pathogenic infection; however, the detailed mechanism by which infection affects the exoskeleton remains largely unknown. Here, we report that the damage to the shrimp exoskeleton is caused by modulation of host lipid profiles after infection with white spot syndrome virus (WSSV). WSSV infection disrupts the mechanical performance of the exoskeleton by inducing the expression of a chitinase (Chi2) in the sub-cuticle epidermis and decreasing the cuticle chitin content. The induction of Chi2 expression is mediated by a nuclear receptor that can be activated by certain enriched long-chain saturated fatty acids after infection. The damage to the exoskeleton, an aftereffect of the induction of host lipogenesis by WSSV, significantly impairs the motor ability of shrimp. Blocking the WSSV-caused lipogenesis restored the mechanical performance of the cuticle and improved the motor ability of infected shrimp. Therefore, this study reveals a mechanism by which WSSV infection modulates shrimp internal metabolism resulting in phenotypic impairment, and provides new insights into the interactions between the arthropod host and virus.

## Author summary

Loosening and softening of the exoskeleton of shrimp is frequently observed after white spot syndrome virus (WSSV) infection. This study reveals that the impairment of the exoskeleton is due to the enrichment of certain long-chain fatty acids (LCFAs), induction of chitinase, and ultimately decrease in cuticle chitin after WSSV infection. The exoskeleton allows for movement through contracting connected muscle; therefore, these WSSV-induced changes impair the motor ability of infected shrimp. The impaired motor ability may increase the chance of an infected shrimp being consumed by a healthy shrimp, thereby facilitating the natural transmission of WSSV. As WSSV exploits LCFAs for virion

no role in the design and conduct of the study, in the collection, analysis, and interpretation of the data, and in the preparation, review, or approval of the manuscript.

**Competing interests:** The authors have declared that no competing interests exist.

morphogenesis, we also show that blocking WSSV-induced lipogenesis restores exoskeleton performance, providing a possible strategy for the inhibition of viral transmission.

## Introduction

A prominent feature of arthropods is the exoskeleton, which covers the whole body. The relatively rigid exoskeleton protects internal tissues, supports the soft body, and contributes to locomotion. Therefore, the exoskeleton is vital for the survival, adaption, and prosperity of arthropods [1]. The exoskeleton consists of several layers of cuticle. The outer thin layer is the non-chitinous epicuticle, which contains hydrophobic lipoproteins and lipids and functions as a barrier. The inner thick layer is the chitinous procuticle, which is composed of microfibers of chitin embedded in a matrix of protein; it can be further divided into the inner endocuticle and outer exocuticle. The cross-bonding of the chitin-protein complex in this layer confers strength and hardness to the skeletal material. In crustaceans, mineral deposition usually occurs in the chitinous procuticle between and within chitin-protein fibrils, making the crustacean cuticle mechanically rigid. The innermost layer of the arthropod exoskeleton is the epidermis and the supportive basement membrane. The membranous layer supports secretory epidermal cells that produce all overlying layers [2].

The major organic component and the core organizer for the arthropod procuticle, which makes up the bulk of the cuticle, is chitin [3]. Chitin is a polymer of β-1,4-linked $N$-acetyl-D-glucosamine (GlcNAc). By cross-linking the cuticular proteins, chitin plays a dominant role in cuticle assembly. Chitin-protein complex fibers are organized in parallel, forming numerous layered flat sheets, which can be arranged into a helicoidal architecture. The chitin arrangement determines the mechanical properties of the cuticle [4,5]. During cuticle formation, chitin is synthesized in the apical epithelial cell membrane by membrane-integrated chitin synthase, which also functions to extrude the chitin fibers into the extracellular space [6]. With the continuous increase in body length and weight, the cuticle would limit the growth of arthropods, especially crustaceans. To overcome this limitation conferred by encasement in a rigid covering, arthropods have to molt to shed the old exoskeleton for growth [7]. During this process, epidermal cells release chitin degradation enzymes, termed chitinases, which are responsible for the hydrolysis and decomposition of chitin during molting [8]. Therefore, the biosynthesis and recycling of chitin and the cuticle must be tightly regulated. Disruption of chitin metabolism would lead to an impaired exoskeleton or even death in insects and crustaceans [9–12].

Shrimp aquaculture produces a high-quality protein source and is important for food and nutritional security worldwide [13]. However, the shrimp industry has long been hindered by various pathogens. Loosening and softening of the shrimp exoskeleton is a common pathological characteristic of many diseases. For example, white spot syndrome virus (WSSV) is the most pathogenic and devastating viral agent affecting the shrimp industry [14]. The cuticular epidermis is a primary target tissue and the most severely damaged tissue after WSSV infection [15]. A typical clinical sign of WSSV infection in shrimp is a loose and soft cuticle. However, the detailed mechanism underlying exoskeleton damage remains largely unknown. Revealing this mechanism would be helpful to develop the strategies to control and cure shrimp diseases.

This study aimed to reveal how WSSV infection impairs the exoskeleton, using kuruma shrimp (*Marsupenaeus japonicus*) as a model. We showed that the damage is caused by an increase in chitinase expression in the epidermis and decrease in chitin content in the cuticle.

WSSV infection altered lipid metabolism and caused the induction of certain saturated long-chain fatty acids (LCFAs). These saturated LCFAs acted as ligands for the nuclear receptor E75, which is essential for the abnormal induction of chitinases. Blocking the abnormal generation of LCFAs after WSSV infection improved the mechanical performance of the shrimp exoskeleton. Therefore, our study uncovered a mechanism by which the alteration of shrimp lipid metabolism by WSSV damages host exoskeleton integrity, providing new insights into the host-virus interaction.

## Results

### WSSV infection disrupts the mechanical performance of the shrimp exoskeleton by decreasing the cuticle chitin content

The loosening and softening of the shrimp cuticle are typical pathological characteristics of WSSV infection. To reveal the damage to the cuticle caused by WSSV infection on the microscale, scanning electron microscopy (SEM) was used to analyze the cuticle morphology at 72 h after WSSV infection. The SEM images showed obvious morphological changes to the procuticle layer. As shown in Fig 1A, the helicoidally arranged chitin-protein fibers exhibited a lamellar appearance. In the endocuticle of the WSSV-infected shrimp, the integrity of the lamella was damaged, and each lamella appeared scattered (red arrow), with some peeling but attached fragments (yellow arrow), suggesting that chitin-protein fibers were disrupted. To quantitatively characterize the mechanical performance of the cuticle, the stiffness was detected using a texture analyzer. As shown in Fig 1B, the cuticle of heathy shrimp could tolerate a force of approximately 600 g, while WSSV infection lowered the tolerance to approximately 400 g. These data indicated that the integrity of the shrimp cuticle was affected by WSSV infection.

Chitin is the central scaffold of the lamellar structure, accounting for the majority of the organic materials of the cuticle; therefore, we used Raman spectrometry to detect whether the chitin abundance in the shrimp cuticle after WSSV infection. As shown in Fig 1C, according to previous studies, the peak from 2849 $cm^{-1}$ to 2941 $cm^{-1}$ represents chitin [16]. The relative intensity of this peak in the cuticle from WSSV-infected shrimp was lower than that from the healthy shrimp, suggesting a loss of chitin content. To visualize the chitin content in the cuticle, a chitin probe that reacted specifically to chitinous materials, resulting in a green fluorescence signal, was used to stain the cuticle section. The fluorescence signal in the WSSV-infected sample was much weaker than that in the control sample, confirming the decrease in chitin in the cuticle after WSSV infection (Fig 1D). We also evaluated the chitin density in the cuticle. A piece of cuticle was collected to determine its dimension on a coordinate paper. After that, pure chitin was extracted to determine its weight. As shown in Fig 1E, the chitin content per unit area of the cuticle was significantly reduced by WSSV infection.

The exoskeleton provides a surface for muscle attachment and allows for movement through contracting the connected muscle [17]; therefore, we speculated that the disruption in the mechanical performance of the exoskeleton might affect shrimp movement. As shown in Fig 1F and 1G, WSSV infection suppressed the motor ability of shrimp significantly. The fitting movement trajectory of the WSSV-infected shrimp after prodding (S1 Video), which may reflect the ability to evade predation, was much simpler (Fig 1F) and shorter (Fig 1G) than that of the control shrimp. Collectively, these results demonstrated that WSSV infection decreased the abundance of chitin, which is critical for the mechanical performance of the shrimp exoskeleton and shrimp motor ability.

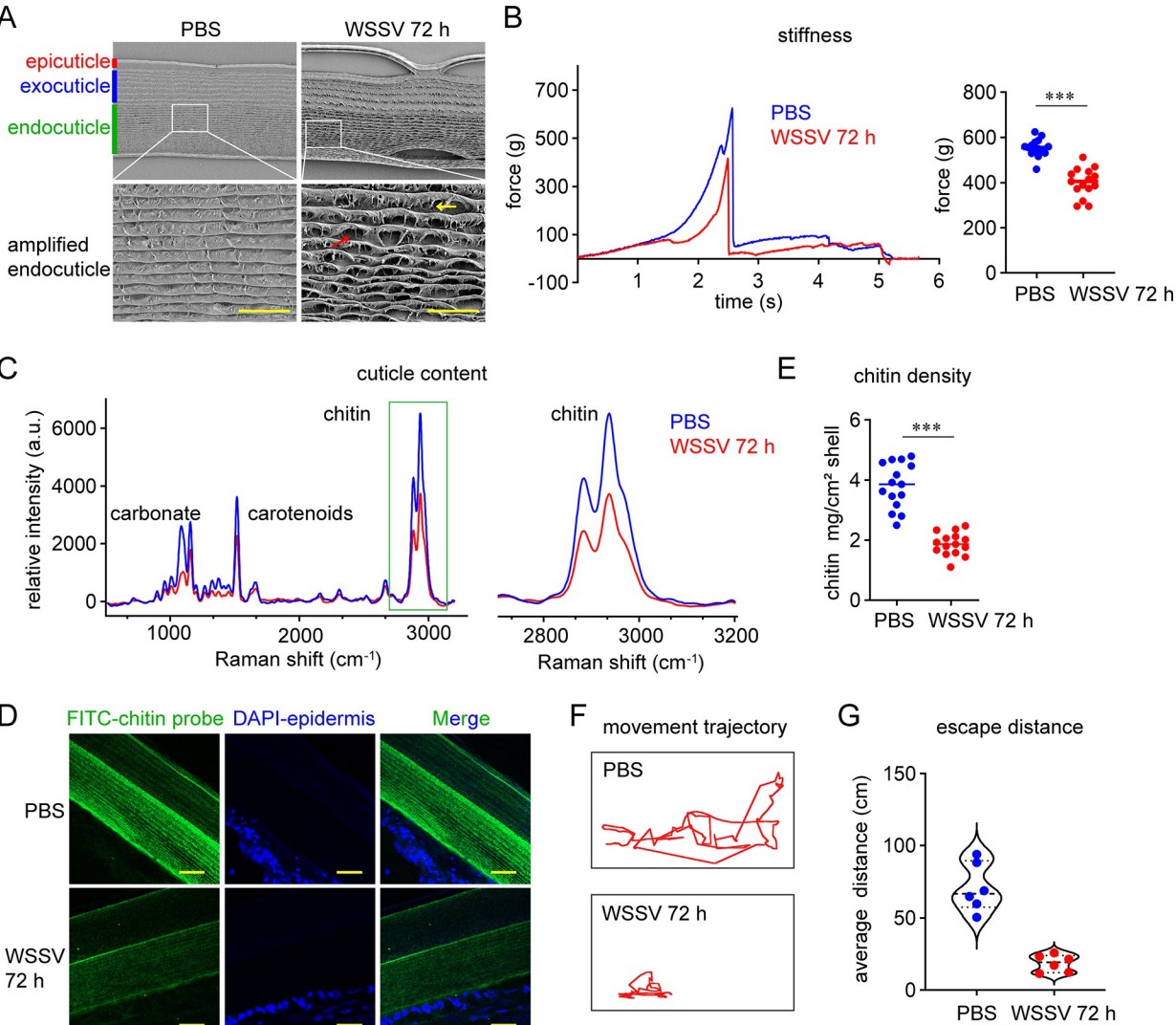

**Fig 1. WSSV infection damages the mechanical performance of the shrimp cuticle and decreases the chitin content.** (A) Disruption of the lamella integrity of the shrimp cuticle, as shown by SEM image. The red arrow indicates the scattered lamella, while the yellow arrow indicates the peeling fragment. Scale bar, 10 μm. (B) Decrease in stiffness after WSSV infection, as shown by a texture analysis with the P/2N probe. Left panel, texture profile of a representative test; right panel, decrease in cuticle stiffness. n = 15 shrimp; the line shows the median (unpaired Student's *t*-test, ***$p < 0.001$). (C) Decrease in the chitin content in the shrimp cuticle after WSSV infection, as shown by Raman characterization. Left panel, Raman spectra (excitation source laser at 532 nm) in the range of 0–3200 cm$^{-1}$ displaying the major characteristic peaks of carbonate, carotenoids, and chitin; right panel, Raman spectra focused on the peaks corresponding to chitin. (D) Decrease in the chitin content in the shrimp cuticle after WSSV infection, as determined by an immunohistochemical analysis. The shrimp cuticle was collected for frozen sectioning. The recombinant chitin binding domain (CBD) from *Bacillus circulans* WL-12 Chitinase A1 was used to mark chitin, and FITC-labeled antibodies were used to visualize the bound CBD; scale bar, 20 μm. (E) Decrease in the chitin density in the shrimp cuticle after WSSV infection. The dimension of shrimp cuticle was determined on the coordinate paper. The cuticle was treated by boiling in 10% NaOH and 3.6% HCl to remove the protein and carbonate. The obtained chitin was dried and weighed. n = 15 shrimp. The line shows the median; unpaired Student's *t*-test, ***$p < 0.001$. (F–G) Impairment in shrimp motor ability after WSSV infection. Shrimp were infected with WSSV. After 72 h, the shrimp were transferred into a new tank, behind which a coordinate paper was placed. The swimming trajectory of the shrimp after a slight prod was recorded using a camera, and the video was analyzed using the Anima Tracker plugin in ImageJ to fit the movement trajectory (F) and to determine the escape distance (G). n = 6 shrimp. The image of movement trajectory was representative of these independent replicates. SEM images, Raman spectra, and immunohistochemical figures are representative of three independent replicates.

## WSSV infection induces chitinase expression to decrease chitin in the shrimp cuticle

Chitinase is the major hydrolytic enzyme for chitin. As shown in Fig 2A, chitinase activity in the sub-cuticle epidermis was significantly higher in the WSSV-infected sample than in the control sample. To determine the enzyme responsible for WSSV-mediated chitin degradation, we evaluated the expression profiles of all 18 chitinases identified in the kuruma shrimp genome (GenBank GCA_017312705.1). *Chi2* and *Chi13* were detected in the sub-cuticle epidermis and increased in response to WSSV, suggesting their possible involvement in chitin decrease (S1 Fig). *Chi2* and *Chi13* expression levels increased significantly at the late stage (24 h post infection) of infection (S2 Fig). Thereafter, RNA interference (RNAi) was carried out to silence their expression (S3 Fig) to verify their possible involvement in WSSV-induced chitin degradation. The results showed that knockdown of *Chi2* attenuated the increase in chitinase activity (Fig 2B) and relieved the decrease in the chitin peak in the Raman spectrum caused by

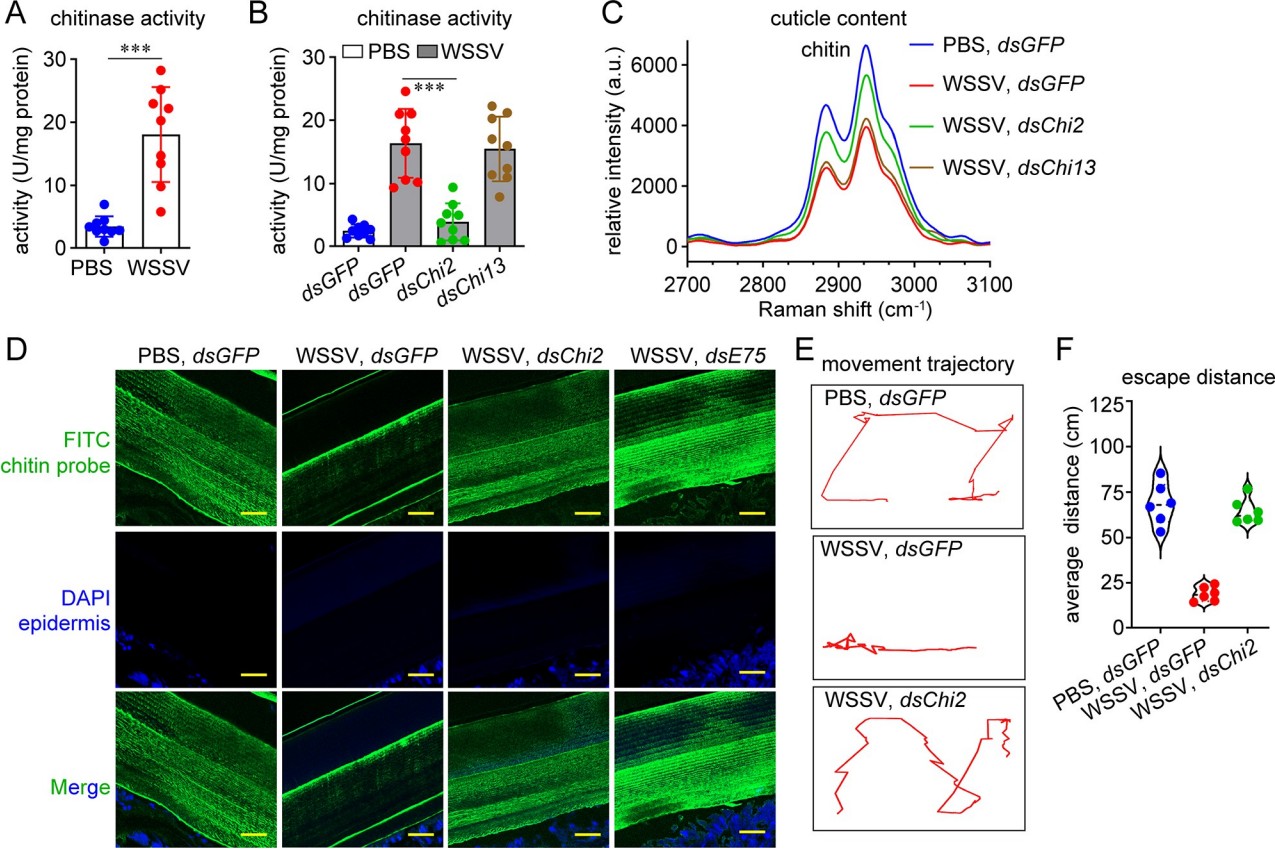

**Fig 2. WSSV infection induces *Chi2* expression and thereby decreases chitin.** (A) Increase in chitinase activity in the sub-cuticle epidermis after WSSV infection. The sub-cuticular tissue was collected and homogenized in PBS to obtain the supernatant. Chitinase activity was determined by monitoring the ability of tissue homogenate to convert colloidal chitin to *N*-acetyl glucosamine. n = 9 shrimp. Mean ± SD (unpaired Student's t-*test*, ***$p < 0.001$). (B) Inhibition of the WSSV-induced chitinase increase by *Chi2* knockdown. RNAi was performed at 24 h after WSSV infection. Chitinase activity in the sub-cuticle epidermis was determined another 48 later. n = 9 shrimp. Mean ± SD (unpaired Student's *t*-test, ***$p < 0.001$). (C) Inhibition of the WSSV-induced chitin decrease by *Chi2* knockdown, as analyzed using Raman spectroscopy. RNAi was performed at 24 h after WSSV infection. The cuticle was collected for he Raman analysis (excitation at 532 nm) after 48 h. (D) Inhibition of the WSSV-induced chitin decrease by *Chi2* knockdown, as analyzed using an immunohistochemical analysis. Scale bar, 20 μm. (E–F) Improvement in shrimp motor ability after WSSV infection with *Chi2* knockdown. Shrimp were infected with WSSV and injected with dsRNA 24 h later. The movement trajectory(E) and escape distance (F) were determined after 48 h. n = 6 shrimp. The image of movement trajectory was representative of these independent replicates. Images of Raman spectra and immunohistochemical analysis are representative of three independent replicates.

WSSV infection (Fig 2C), while the knockdown of *Chi13* had almost no effect. Considering that Chi2 is a group II chitinase consisting of multiple glycoside hydrolase family 18 catalytic domains (S1 Fig) and is essential for chitin degradation in the arthropod molting process [12,18], the indispensability of Chi2 in the chitin decrease after WSSV infection is reasonable. We also proved that *Chi2* knockdown could restore the cuticle chitin content to levels in uninfected individuals by using the immunofluorescence analysis (Fig 2D). In addition, *Chi2* knockdown restored the shrimp's motor ability (Fig 2E and 2F). These data suggested that *Chi2* induction after WSSV infection contributes to the decrease in the chitin content in the exoskeleton and therefore to the impairment of shrimp motor ability.

## E75 is essential for the abnormal induction of *Chi2* after WSSV infection

Next, we evaluated how WSSV infection led to the induction of *Chi2* expression. Chitinases in the epidermis are mainly responsible for chitin recycling and cuticle biosynthesis during molting and development and are controlled by ecdysone signaling [19]. The presence of potential binding sites for almost all essential transcription factors for ecdysone signaling in the promoter sequences of *Chi2* suggested that WSSV infection might interfere with the regulation of chitinase expression at some node of ecdysone signaling (Fig 3A). Therefore, we tested whether any transcription factor in the ecdysone signaling pathway was related to the abnormal induction of *Chi2*. As shown in Fig 3B, although the knockdown of several factors, including retinoid X receptor (RXR), ecdysone-induced protein 93 (E93), hormone receptor 4 (HR4), and fushi tarazu-F1 (FTZ-F1), had inhibitory effects on the induction of *Chi2*, only the knockdown of *E75* completely abrogated the induction, suggesting that E75 might be an essential node. To confirm the role of E75, chitin abundance in the shrimp cuticle was detected after WSSV infection and *E75* knockdown. The results showed that *E75* knockdown could completely inhibit the decrease in chitin abundance (Figs 3C and 2D). Moreover, the disruption in the motor ability of shrimp after WSSV infection was also restored by *E75* knockdown (Fig 3D and 3E).

Afterwards, an electrophoretic mobility shift assay (EMSA) was performed to determine whether E75 interacts with predicted E75-binding sites in the *Chi2* promoter sequence (Fig 3F). The results revealed that the DNA binding domain of E75 could bind to the wild-type probe. The binding was also confirmed by competition assays (Fig 3G). Next, a chromatin immunoprecipitation (ChIP) assay was performed to further confirm the interaction between E75 and the promoter of *Chi2*. As shown in Fig 3H, a positive signal was detected from the immunoprecipitates of E75 antibodies only when using the WSSV-infected epidermis as the pool for ChIP. Therefore, the above data suggested that E75 transcriptionally regulated *Chi2* expression directly in the presence of WSSV infection.

## Saturated LCFAs are ligands that activate the nuclear receptor E75 for *Chi2* induction

Next, we investigated how WSSV infection potentiated E75-mediated *Chi2* induction. E75 belongs to the nuclear receptor family. Nuclear receptors are ligand-gated transcription factors. Their transcriptional activity is usually controlled through the interaction of a ligand binding domain (LBD) with certain small and fat-soluble ligands. Ligand binding acts as the molecular switch to turn the receptors into transcriptional activators [20,21]. Previous studies have demonstrated that many endogenous metabolites could activate nuclear receptors [22,23]. As E75 expression was not influenced by WSSV infection (S2 Fig), we supposed that metabolites enriched after WSSV infection would act as ligands for E75. Therefore, liquid chromatography mass spectrometry (LC-MS) and gas chromatography mass spectrometry (GC-MS) approaches

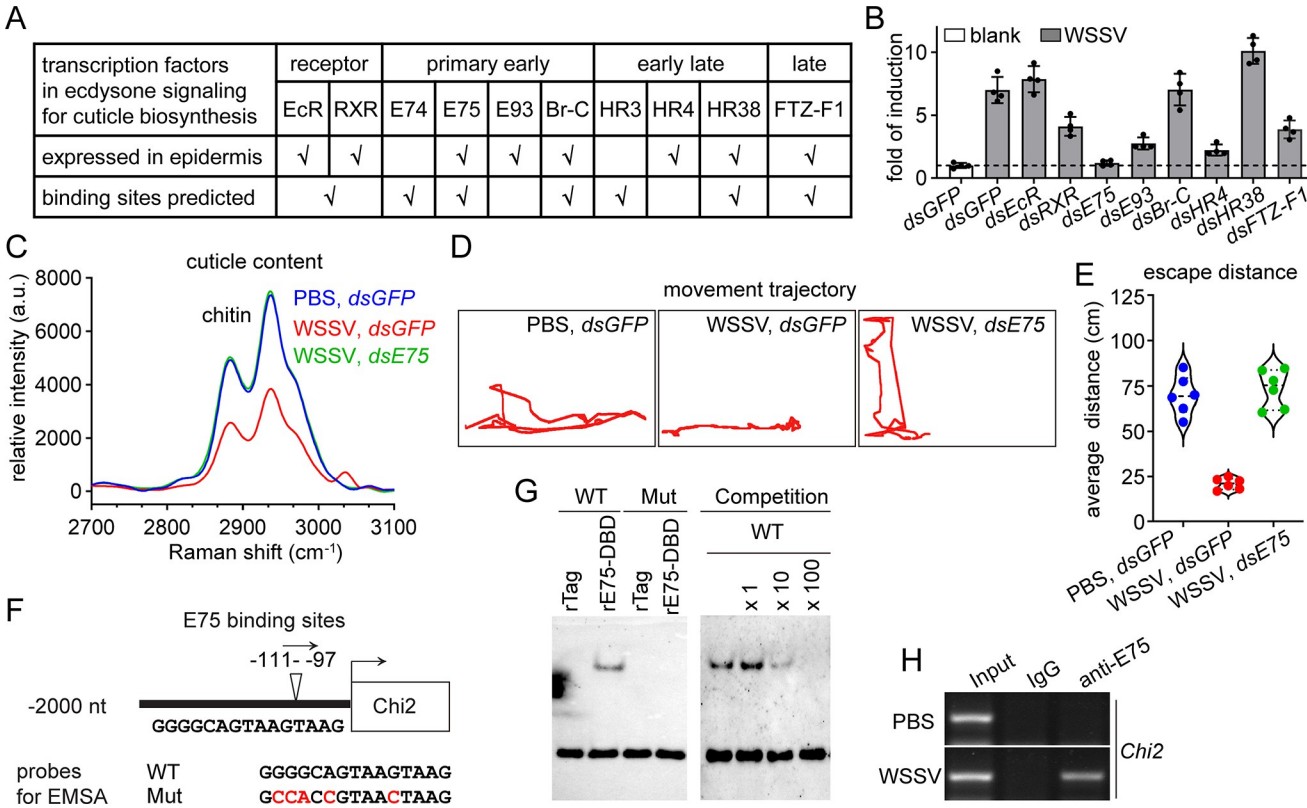

**Fig 3. WSSV infection induces *Chi2* expression through E75.** (A) Sequential transcription factors essential for cuticle biosynthesis. The expression of these factors in the epidermis was determined using RT-PCR. The 5′-untranslated regions of *Chi2* were obtained from the shrimp genome (GenBank GCA_017312705.1) and potential transcription factor binding sites were predicted using the online tool JASPAR. (B) Effect of the knockdown of selected candidate transcription factors on WSSV-caused *Chi2* induction. RNAi was performed at 24 h after WSSV infection. *Chi2* expression was detected after 24 h. Results are presented as the mean ± SD from three replicates. (C) Inhibition of the WSSV-caused chitin decrease by *E75* knockdown, as analyzed by Raman spectroscopy (excitation at 532 nm). RNAi was performed at 24 h after WSSV infection. The cuticle was collected for the Raman analysis after another 48 h. (D–E) Improvement in shrimp motor ability after WSSV infection with *E75* knockdown. Shrimp were infected with WSSV and injected with dsRNA 24 h later. After another 48 h, the shrimp were transferred into new tank to trace their movement trajectory (D) and to determine the escape distance (E) after slight prodding. n = 6 shrimp. The image of movement trajectory was representative of these independent replicates. (F) Illustration of the E75 binding sites in the promoter of *Chi2*. (G) Interaction of recombinant E75 with biotin-labeled oligonucleotides encoding the E75-binding site, as analyzed using EMSA. Competition assays were performed in the presence of excess unlabeled oligonucleotides, as indicated. (H) Binding of E75 to the promoter fragments of *Chi2* after WSSV infection, as analyzed using a ChIP assay. The sub-cuticle epidermis was collected at 48 h after WSSV infection as a pool for the ChIP assay. Images of Raman spectra, EMSA, and ChIP analysis are representative of three independent replicates.

were used in combination for the metabolic profiling of the epidermis to identify differential metabolites caused by WSSV infection. A partial least-squares discriminant analysis revealed clear separation of the epidermis metabolite profiles between the WSSV-infection and control groups (S4 Fig). Among 20 enriched metabolites (Fig 4A), special attention was paid to myristic acid (MA, C14:0), a saturated LCFA, because LCFAs are typical ligands for nuclear receptors [24,25]. Moreover, several saturated LCFAs, including MA, palmitic acid (PA, C16:0), and stearic acid (SA, C18:0), act as ligands for peroxisome proliferator activator receptors (PPARs), which are the closest vertebrate relatives of E75 [26–28]. Interestingly, although both PA and SA were not filtered out in the screening owing to the strict cut-off, their abundance indeed increased significantly by WSSV infection in the shrimp epidermis (Fig 4B).

Next, we studied whether the increases in these saturated LCFAs contributed to the induction of chitinase expression. As demonstrated in Fig 4C, MA, PA, or SA could induce the expression of *Chi2 in vivo*. Moreover, MA increased chitinase activity in the epidermis (Fig 4D)

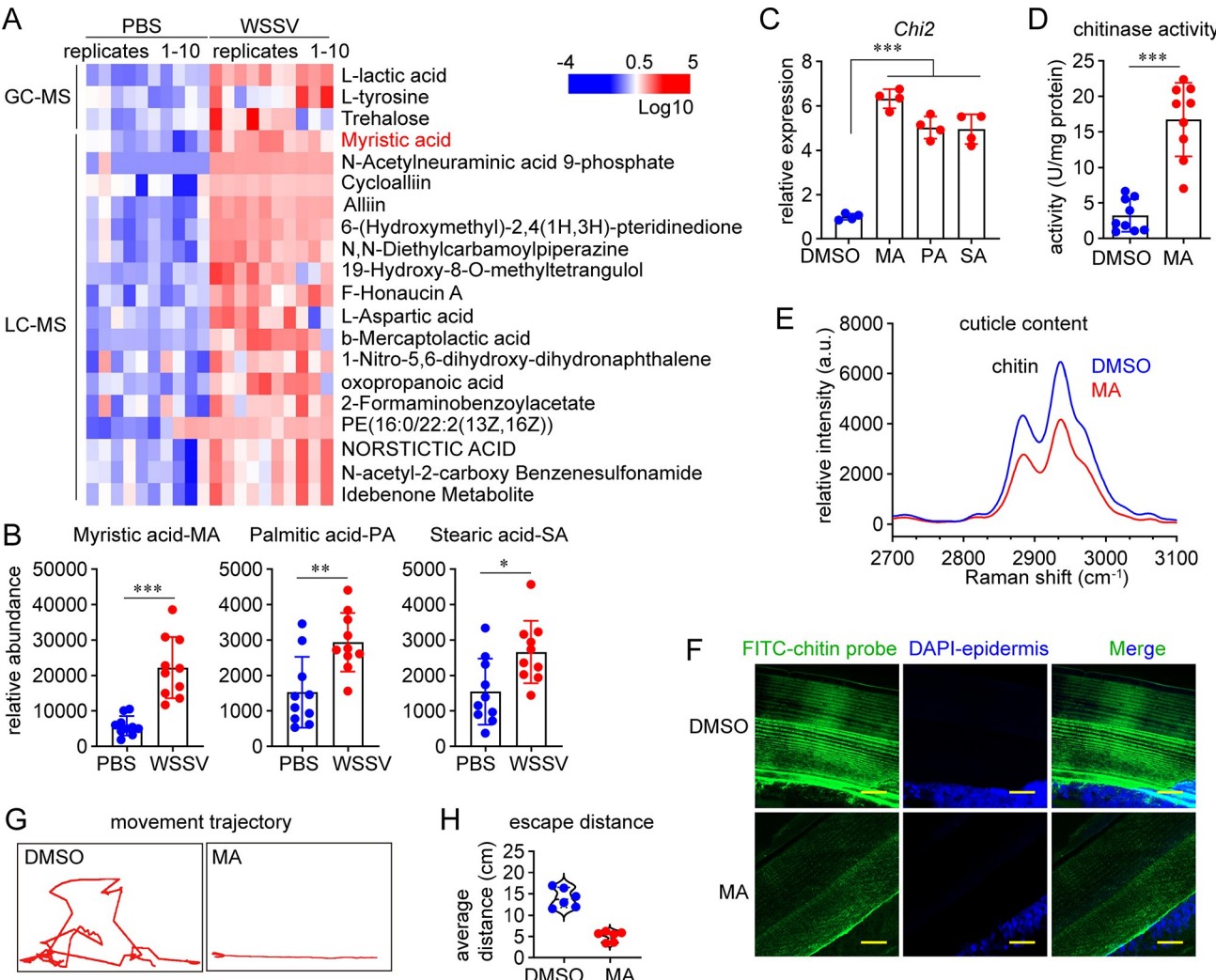

**Fig 4. WSSV-induced saturated LCFAs activate *Chi2* expression.** (A) Increase in the relative abundance of differential metabolites in the epidermis, as shown using a heatmap. The sub-cuticle epidermis was collected at 24 h after WSSV infection or PBS injection for the metabolomics analysis. The color bar indicates the gradient of normalized abundance. Differential metabolites were identified with VIP > 1.0, $p < 0.05$, and fold change > 2. Each group consisted of 10 independent replicates. (B) Increase in the relative abundance of three saturated LCFAs after WSSV infection. Mean ± SD, unpaired Student's *t*-test, *$p < 0.05$, **$p < 0.01$, ***$p < 0.001$. (C) Induction of *Chi2* by three saturated LCFAs. The saturated LCFAs were administered to the shrimp hemocoel at 5 μg per shrimp. DMSO was used as a control. *Chi2* expression was detected 12 h later using qRT-PCR. Mean ± SD, unpaired Student's *t*-test, ***$p < 0.001$. (D) Induction of chitinase activity by MA in the sub-cuticular epidermis. Chitinase activity was detected at 24 h after MA administration. n = 9 shrimp. Mean ± SD, unpaired Student's *t*-test, ***$p < 0.001$. (E) Decrease in the cuticle chitin content after MA application, as analyzed using Raman spectroscopy (excitation at 532 nm). The cuticle was collected for the Raman analysis at 48 h after MA injection. (F) Decrease in the cuticle chitin content after MA application, as analyzed using an immunohistochemical analysis. Scale bar, 20 μm. (G–H) Impairment of shrimp motor ability by MA application. Shrimp were transferred into a new tank at 48 h after MA administration to trace their movement trajectory(G) and to determine the escape distance (H) after slight prodding. n = 6 shrimp. The image of movement trajectory was representative of these independent replicates. Images of Raman characterization and immunofluorescent assay are representative of three independent replicates.

and decreased the chitin content in the cuticle (Fig 4E and 4F). Interestingly, MA application also significantly impaired the motor ability of shrimp (Fig 4G and 4H). Together, these data showed that saturated LCFAs were induced by WSSV infection and were related to *Chi2* expression.

To determine the necessity of E75 for LCFA-induced *Chi2* expression, a ChIP assay was performed after the application of MA. The fragments containing E75 binding sites were

present in the immunoprecipitates from the epidermis of MA-treated shrimp (Fig 5A). These data suggested that MA could modulate E75-mediated *Chi2* transcription. Next, E75 expression was silenced to check whether the activity of MA was affected. As shown in Fig 5B, *E75* pre-knockdown inhibited the MA-induced increase in *Chi2*. Moreover, the increase in epidermis chitinase activity (Fig 5C) and decrease in cuticle chitin content (Fig 5D) caused by MA application were blocked when E75 expression was pre-silenced. These results confirmed the presence of the LCFA-E75-*Chi2* cascade in shrimp.

To characterize the interaction between E75 and the saturated LCFAs, molecular docking was performed. As shown in Fig 5E, the LBD of E75 was predicted to have a typical structure of the E domain present in nuclear receptors [29]. The α-helices that fold into an alpha helical sandwich created a putative ligand-binding pocket, which could accommodate each of the three fatty acids. The interaction of the E75 LBD and MA was further validated using an

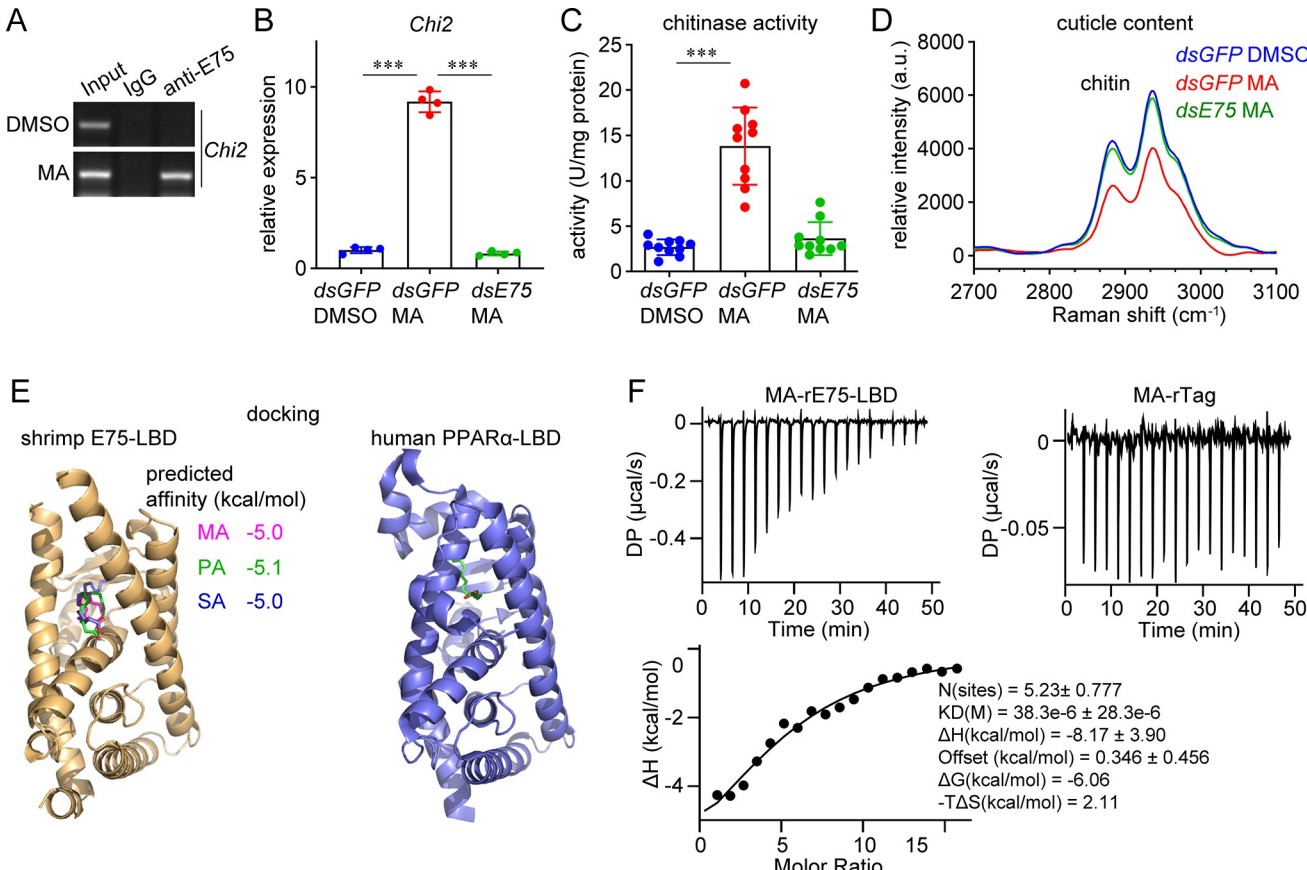

**Fig 5. The LCFA MA acts as a ligand for E75 to induce *Chi2* expression.** (A) Induction of the E75-mediated transcriptional regulation of *Chi2* by MA. A ChIP assay was performed at 12 h after MA (5 μg) or DMSO administration. (B) Inhibition of the MA-induced *Chi2* expression by *E75* knockdown. MA was administered to shrimp after *E75* silencing and the expression of *Chi2* was detected 12 h later using qRT-PCR. Mean ± SD, unpaired Student's *t*-test, ***$p < 0.001$. (C) Inhibition of the MA-induced increase in sub-cuticular epidermis chitinase activity by *E75* knockdown. MA was administered to shrimp after *E75* silencing and chitinase activity was determined 24 h later. Mean ± SD, unpaired Student's *t*-test, ***$p < 0.001$. (D) Inhibition of the MA-induced decrease in the cuticle chitin content by *E75* knockdown. MA was administered to shrimp after *E75* silencing. The cuticle was collected for Raman spectroscopy (excitation at 532 nm) at 48 h after MA injection. (E) Molecular docking of the interaction between E75 LBD and three saturated LCFAs. The modeled structures of the complexes formed by E75-LBD and each fatty acid were overlaid for display. The experimentally determined structure of human PPARα-PA (PDB: 6KAX) is shown as a reference. The structure of E75 LBD was predicted using AlphaFold2. Docking was performed using AutoDock Vina 1.1.2, and the results were visualized using PyMOL version 2.4.1. (F) Characterization of the interaction between MA and rE75-LBD using an ITC assay. MA (800 μM) in the syringe was injected into cells containing rE75-LBD or the control rTag (10 μM) and stirred at 750 rpm for 150 s; 18 injections were performed. Raman spectra and immunofluorescent and ITC assays results are representative of three independent replicates.

isothermal titration calorimetry assay. The results showed that the interaction mainly fitted the one set of sites model, and the $K_D$ was approximately 36 μM (Fig 5F). Therefore, the increased abundance of saturated LCFAs after WSSV infection induced chitinase expression, thereby decreasing the chitin content in the cuticle and impairing shrimp motor ability.

## Blocking WSSV-caused lipogenesis inhibited the chitin decrease and improved exoskeleton performance

Next, we used the fatty acid synthase inhibitor C75 to block lipogenesis to determine whether the disruption of shrimp cuticle integrity would be relieved. The application of C75 suppressed the induction of *Chi2* (Fig 6A) and the increase of chitinase activity in the epidermis (Fig 6B) after WSSV infection. The decrease in chitin abundance caused by WSSV infection was also inhibited by C75, as revealed by Raman spectroscopy (Fig 6C), chitin fluorescent signals (Fig 6D), and the measurement of chitin density (Fig 6E). Moreover, the damage to the cuticle morphology and the change in the mechanical properties of the cuticle caused by WSSV infection were also abrogated. As shown in Fig 6F, the scattering of the chitin-protein lamella caused by WSSV infection disappeared, and the chitin-protein fibers were restored to a normal appearance after C75 application. Furthermore, the decrease in cuticle stiffness caused by WSSV infection was also attenuated by C75 (Fig 6G). Collectively, these results supported the view that blocking WSSV-caused lipogenesis could prevent the damage to the shrimp exoskeleton mechanical performance after WSSV infection. We also evaluated whether lipogenesis inhibition improves the impaired motor ability caused by WSSV infection. As shown in Fig 6H and 6I, the fitting movement trajectory of the WSSV/C75 shrimp after prodding was more complicated (Fig 6H) and longer (Fig 6I) than that of the WSSV/Ctrl shrimp, suggesting that C75 application improved the motor ability of the shrimp.

## Discussion

In addition to functioning as a surface barrier to protect internal organs, the arthropod exoskeleton consists of multiple joined plates and attaches to muscle, thus allowing for complex movement [17]. Therefore, damage to the shrimp exoskeleton after WSSV infection would significantly impair movement, as evidenced in Fig 1F and 1G. The consumption of dead or feeble WSSV-infected shrimp by healthy shrimp is the major natural transmission route of WSSV [30]. The impaired movement ability might increase the chance of an infected shrimp being consumed by a healthy shrimp. Therefore, the damage to the shrimp exoskeleton might make a considerable contribution to the natural transmission of WSSV, the most severe pathogen for cultured shrimp with an extremely high transmission rate, in aquaculture farms [30]. In this study, we proved that the decrease in the chitin content, the major organic material of the shrimp exoskeleton, was due to increased chitinase expression, and that the induction of chitinases was an additional effect caused by the disruption of host lipid metabolism. These results clarified the mechanisms underlying shrimp exoskeleton damage after WSSV infection. By blocking WSSV-induced lipogenesis, we successfully restored the chitin content and the mechanical properties of the shrimp exoskeleton. Moreover, the impaired movement after WSSV infection was restored by application of a fatty acid synthesis inhibitor. This suggested the potential to develop an approach to inhibit the transmission of WSSV in farms by interfering with virus-manipulated lipid metabolism. Moreover, identification of chemicals able to compete with LCFAs for binding to E75 may be another feasible alternative to block the induction of chitinases and suppress WSSV transmission.

Chitinases (EC 3.2.1.14) belong to the glycosyl hydrolase family and are a group of hydrolytic enzymes possessing a chitin-binding domain and catalytic domain, which can cleave the

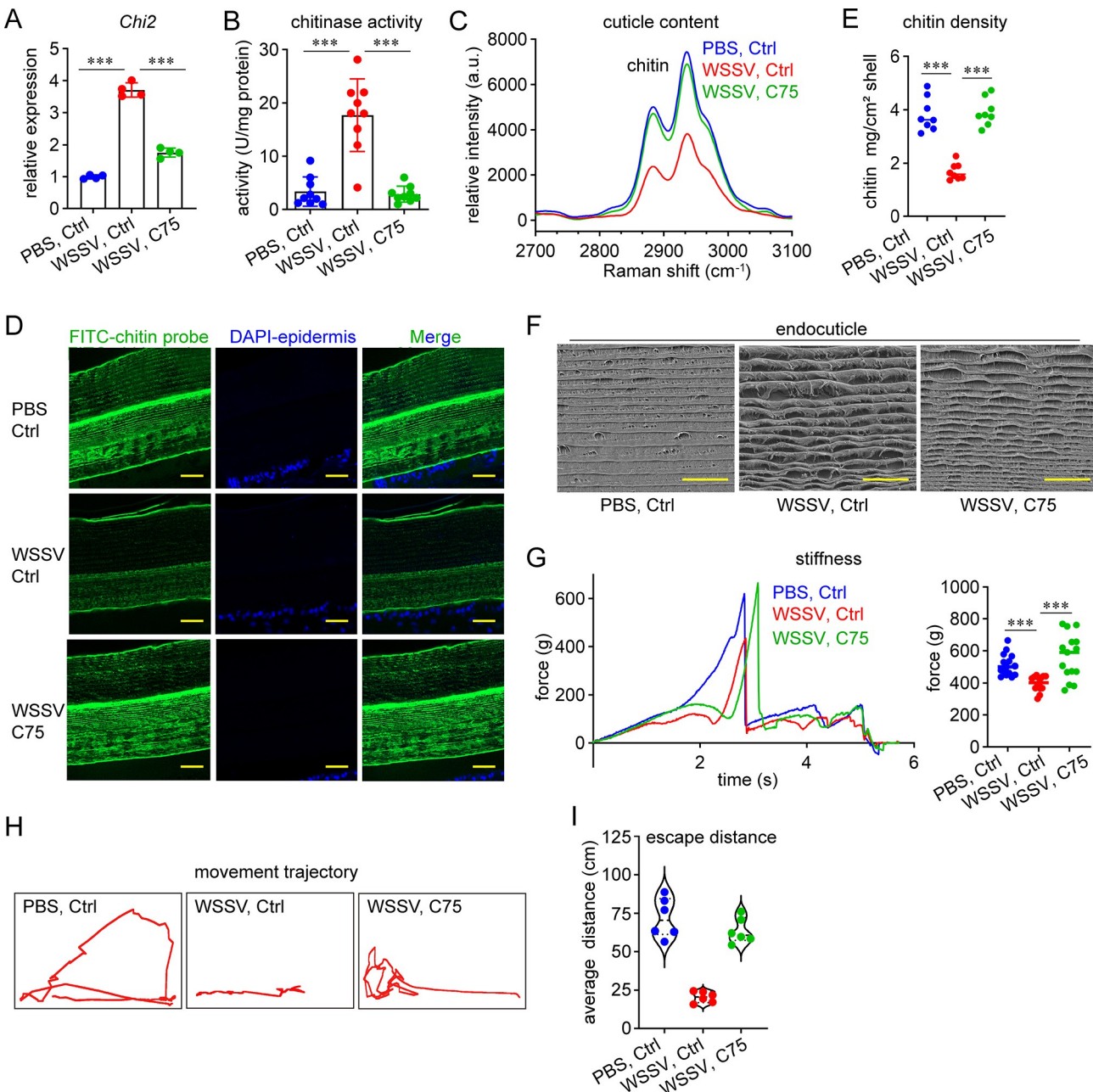

**Fig 6. Blocking of WSSV-caused lipogenesis restores the chitin content and mechanical performance of the shrimp cuticle.** (A) Inhibition of WSSV-caused *Chi2* induction by C75 application. C75 was administered to the shrimp hemocoel (5 μg) at 12 h after WSSV infection. *Chi2* expression was detected another 24 h later. Mean ± SD, unpaired Student's *t*-test, ***$p < 0.001$. (B) Inhibition of the WSSV-caused increase in chitinase activity by C75 application in the sub-cuticular epidermis. C75 was administered into shrimp at 12 h after WSSV infection. Chitinase activity was detected another 36 h later. n = 9 shrimp. Mean ± SD, unpaired Student's *t*-test, ***$p < 0.001$. (C) Inhibition of the WSSV-caused chitin decrease by C75 application, as analyzed using Raman spectroscopy (excitation at 532 nm). The cuticle was collected at 72 h after WSSV infection and 60 h after C75 application. (D) Inhibition of the WSSV-mediated chitin decrease by C75 application, as analyzed using an immunohistochemical analysis. Scale bar, 20 μm. (E) Inhibition of the WSSV-caused decrease in the chitin density in the shrimp cuticle by C75 application. n = 8 shrimp. The line shows the median. Unpaired Student's *t*-test, ***$p < 0.001$. (F) Restoration of the microstructure of the shrimp cuticle by C75 application, as shown by SEM analysis. Scale bar, 10 μm. (G) Restoring the stiffness of the shrimp cuticle by C75 application, as shown by a texture analysis. Left panel, texture profile of a representative test; right panel, decrease in cuticle stiffness. n = 15 shrimp; the line shows the median (unpaired Student's t-test, ***$p < 0.001$). (H–I) Restoration of shrimp motor ability by C75 application. Shrimp were transferred into a new tank to trace their movement trajectory(H) and to determine the escape distance (I) after slight prodding. n = 6 shrimp. The image of movement trajectory was representative of these independent replicates. SEM images, Raman spectra, and immunohistochemical figures are representative of three independent replicates.

glycosidic bond in chitin. In crustaceans, chitinases are usually expressed in the hepatopancreas and cuticular tissues (integument, blade, and tail fan) [31]. The chitinases expressed in the hepatopancreas are responsible for the digestion of chitin-containing food, and hepatopancreas chitinase activity does not fluctuate significantly during the molting cycle [32]. In contrast, chitinase activity in the cuticular tissues is induced prior to molting. These chitinases dissolve the chitin in the old cuticle into soluble chitooligosaccharides, which can be resorbed for the synthesis of the new exoskeleton; therefore, they play indispensable roles in the molting and growth of crustaceans [33,34]. Herein, the critical chitinase Chi2 was expressed at a relatively high level in cuticle tissues, suggesting its probable involvement in exoskeleton reshaping during periodic molting. Moreover, Chi2 is a member of the unique group II chitinases, which consist of four or five catalytic domains and are the most important chitinases for molting in arthropods [35]. The abnormal induction of Chi2 and damage to the shrimp exoskeleton indicated that WSSV infection might have disrupted the shaping of the shrimp exoskeleton. Interestingly, previous studies have shown that WSSV infection induces the expression of a set of cuticular proteins, in addition to chitinases, which were related to the pathological changes related to WSSV infection [36]. For example, the strong upregulation of calcification-associated peptide-1 (CAP-1) might contribute to the formation of white spots, which are abnormal deposits of calcium salts in the cuticle [37]; the induction of carboxypeptidases and cathepsins might play a role in digesting the sub-cuticular tissue, resulting in loosening of the cuticle [38]. Although it is not clear whether their abnormal induction and that of Chi2 are accomplished in the same manner, the induction of these cuticular-related enzymes and peptides demonstrates that WSSV infection indeed disrupts the shaping of the shrimp exoskeleton.

The shrimp exoskeleton shaping cycle is generally coordinated by variation in ecdysteroid titers [39]. 20-Hydroxyecdysone (20E), the primary active form of ecdysteroid, is synthesized and transported into target cells, where it binds to its receptor, ecdysone receptor (EcR), a member of the nuclear receptor family [40]. EcR heterodimerizes with RXR, binds to hormone response elements, and directs the transcription of 20E primary response genes (including *Br-C*, *E94*, *E75*, *E74*, and *FTZ-F1*) [41]. Many of the primary response genes encode nuclear receptors, which would further regulate the downstream transcriptional response, and induce the expression of secondary molting-related genes. We found that the interruption of shrimp chitin metabolism by WSSV infection was achieved independently of 20E and its receptors. WSSV infection indeed led to an approximately two-fold increase in the 20E titer (S5 Fig). However, only an increase in the ecdysone titer of over 20-fold had been shown to drive the molting cycle [42]. Therefore, the induction of *Chi2* may be not related to the slight variation in 20E titers after WSSV infection. Moreover, the knockdown of *EcR* did not impair virus-mediated *Chi2* induction. Instead, the knockdown of *E75* completely inhibited *Chi2* induction after WSSV infection. Therefore, E75 might be the critical node at which virus infection interferes with chitin metabolism. As a ligand-gated nuclear receptor, E75 participates in arthropod metamorphosis and reproduction [43,44]. Interestingly, previous studies have shown that E75 is involved in chitin recycling and cuticle generation by regulating the expression of chitinases in insects and crustaceans [45], and the knockdown of *E75* results in molting defects [46]. However, different from the finding that E75 regulates exoskeleton synthesis under the control of ecdysone in a physiological context, this study uncovered a novel mechanism by which E75 contributes to exoskeleton damage in a disease context under the control of virus-induced fatty acids.

Fatty acids, derived from nutritional sources or endogenous metabolism, act as ligands and activators for many nuclear receptors. For example, fatty acids could directly interact with PPARα and PPARγ to regulate gene expression [25]. Fatty acids are frequently modified at the head group (glycerol, amino acid, and carbohydrates) or at the hydrophobic tail (saturation and desaturation). These modifications could increase the diversity of this group of ligands

substantially and further expand the spectrum of ligand-receptor interactions [47–49]. Herein, we found that several saturated LCFAs were endogenous ligands of E75, thus enriching our understanding of ligand-nuclear receptor interactions. Actually, the modulation of fatty acid levels is an effective strategy for WSSV replication. To meet the energy demand for viral genome replication at the early stage of infection, WSSV induces lipolysis to utilize LCFAs via β-oxidation. In contrast, lipogenesis is initiated at the late stage of infection when LCFAs are produced for virion morphogenesis [50,51]. By modulating host lipid metabolism, WSSV creates a favorable environment for replication. In this study, we proved that increased levels of saturated LCFAs at the late stage of infection could also act as a signal to activate E75 to induce the expression of host chitinases, which ultimately damage the shrimp exoskeleton and probably facilitate viral transmission. Therefore, from both microscopic and macroscopic perspectives, the modulation of host lipid profiles is essential for WSSV to prevail in the host-virus arm race, and to be the most prevalent and devastating viral pathogen in shrimp aquaculture. According to published data, infection with other pathogens, including the Decapod iridescent virus 1 and *Vibrio parahaemolyticus*, causes the lipid metabolic reprogramming of shrimp by increasing the amounts of a series of fatty acids, suggesting that the alteration of shrimp lipid metabolism may be general after kinds of pathogens infection [52,53]. In particular, palmitic acid which is concerned in this study was found as a significantly enriched metabolite after *Vibrio parahaemolyticus* infection [53]. Therefore, the mechanism revealed in this study may be also present after other pathogenic infections. However, whether this is the case deserves further study to clarify.

In summary, we have demonstrated how virus infection leads to an impaired shrimp exoskeleton. WSSV modulates host lipid profiles by inducing lipogenesis. Increased saturated LCFAs activate E75 to increase the transcription of chitinase genes. The induced chitinase, Chi2, digested chitin in the cuticle and thus damaged the cuticle integrity and shrimp motor ability (Fig 7). These findings provide new insights into the host-virus interaction.

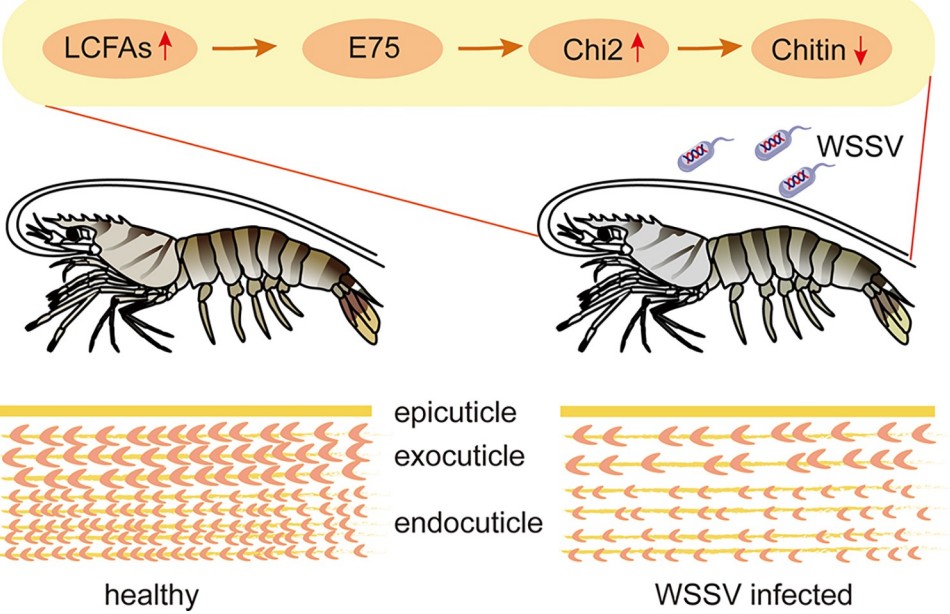

**Fig 7. Working model for this study.** WSSV infection causes an increase in saturated LCFAs, which activate E75 to increase the transcription and production of Chi2. Then, Chi2 digests chitin in the cuticle and thus damages the cuticle integrity.

Interestingly, this study also showed that interference with lipogenesis might suppress virus transmission and highlighted the potential of C75 and other lipogenesis inhibitors for disease control in shrimp aquaculture.

## Materials and methods

### Ethics statement

All animal-related experiments were performed with the approval of the Animal Ethical Committee of Shandong University School of Life Sciences (permit number SYDWLL-2021-98).

### Animals and immune challenge

Healthy intermolt kuruma shrimp (*M. japonicus*), weighing approximately 10 g, were obtained from an aquaculture farm in Jimo, Shandong, China. Shrimp were cultured in aerated seawater at 25˚C, fed daily with a commercial diet, and randomly selected for investigation. The original WSSV strain was a gift from the East China Sea Fisheries Research Institute (Shanghai, China). To prepare successive inoculums, moribund shrimp gills were homogenized in phosphate-buffered saline (PBS) (140 mM NaCl, 2.7 mM KCl, 10 mM $Na_2HPO_4$, 1.8 mM $KH_2PO_4$, pH 7.4) at a ratio of 1:10 (w/v). The homogenate was frozen and thawed twice and centrifuged at $3000 \times g$ for 10 min at 4˚C. The resultant supernatant was filtered through a 0.45 μm filter. Viral titers were determined according to a previously described protocol [54]. For the infection assay, each shrimp was injected with 50 μL of WSSV inoculum intramuscularly ($10^6$ virions). The same volume of PBS was injected as a control.

### Scanning electronic microscopy (SEM)

SEM was used to observe the shrimp cuticle on the microscale. Shrimp cuticles collected from the dorsal side were cut into $3 \times 5$ mm pieces and fixed in Davidson's AFA fixative (30% ethanol, 22% formalin, and 11.5% acetic acid) for 24 h. After washing with PBS thrice, the fixed samples were dehydrated and embedded in paraffin according to conventional procedures. Then, 6 μm sections were obtained using a rotatory microtome and deparaffinized using xylene. The samples were mounted and sputter-coated with platinum using a sputter coater (HITACHI MC1000; Hitachi, Tokyo, Japan). Images were obtained using a field-emission scanning electron microscope (HITACHI Regulus 8100; Tokyo, Japan).

### Measurement of the mechanical properties of the shrimp shell

The stiffness was detected using a TA. XT plus texture analyzer (Stable Micro Systems, Godalming, UK) with a P/2N probe. The shrimp shell was cut into pieces of approximately 1 $cm^2$ for the test. The test speed was set at 1 mm/s until the sample broke completely.

### Chitin content measurement

Raman spectroscopy was performed to characterize the composition of the shrimp cuticle. Raman signals were collected using an inVia Qontor confocal Raman microscope (Renishaw, New Mills, UK) with a $50 \times$ objective lens. A laser at 532 nm was used as an excitation source, and the laser power was 5% of 50 mW. Shrimp cuticles collected from the dorsal side were cut into $5 \times 5$ mm portions for the test. Origin software (OriginLabs, Northampton, MA, USA) was used for data filtering, spectral analyses, and the generation of plots.

 The chitin density in the shrimp cuticle was also quantified. The cuticles were collected to determine their dimensions on a coordinate paper. Afterwards, the cuticle was dried at 60˚C and boiled in 10% NaOH solution for 1 h to remove proteins and treated with 3.6% HCl for 15

min to obtain transparent chitin, which was then dried at 60˚C and weighed. The chitin content per unit area of the cuticle was determined.

## Immunocytochemical analysis

An immunocytochemical assay was performed to visualize the chitinous materials in the shrimp cuticle. The chitin-binding domain (CBD) of *Bacillus circulans* WL-12 Chitinase A1 was recombinantly expressed in bacterial cells using the pET32a (+) vector, purified using affinity chromatography (see below), and used as a probe to bind to chitinous materials. This probe can be visualized by using an anti-His tag antibody and fluorescein isothiocyanate (FITC)-conjugated secondary antibody. The cryosections (6 μm) were treated with the probe solution (0.1 mg/mL) for 12 h. Afterwards, mouse anti-His tag monoclonal antibodies (Abbkine, Wuhan, China; ABT2050; 1:100 dilution) were used to target the recombinant CBD, which was fused with a 6His tag. After washing with PBS, FITC-conjugated goat anti-mouse (Abbkine; A22110; 1:1000 dilution) was added and incubated for 2 h in the dark. Then, 4′,6-diamidino-2-phenylindole (DAPI, AnaSpec, Fremont, CA, USA; AS-83210; 1:1000 dilution) was used to stain the nuclei in the epidermis. Finally, the slides were washed with PBS and observed under a Zeiss LSM 900 confocal microscope (Carl Zeiss, Jena, Germany). The images were analyzed and presented using ZEN software (Zeiss).

## Generation of recombinant proteins and antibodies

The fragments encoding the CBD of *B. circulans* WL-12 Chitinase A1 and kuruma shrimp E75 were amplified using specific primers (S1 Table) and ligated into the pET32a (+) vector. The *Escherichia coli* Rosetta (DE3) strain was used for recombinant expression under induction by 0.5 mM isopropyl-β-D-thiogalactopyranoside at 37˚C for 4 h. Recombinant proteins were purified using ProteinIso Nickel-nitrilotriacetic acid (Ni-NTA) resin (TransGen Biotech, Beijing, China; DP-101). The proteins were dialyzed in PBS and stored at −80˚C. A tag (termed rTag) expressed by the empty vector was used as a control. The protein concentration was determined by using a Bradford protein assay kit (Sangon Biotech, Shanghai, China; C503031).

Recombinant E75 (1 mg/mL, 1.5 mL) was mixed thoroughly with an equal volume of complete Freund's adjuvant (Sigma-Aldrich, St. Louis, MO, USA; F5881), and used to immunize a male New Zealand white rabbit. The immunization was repeated 25 d later using the incomplete adjuvant (Sigma-Aldrich; F5506) rather than complete adjuvant. After detecting the specificity and titer of the antisera at 7 d after the second immunization, the rabbit was sacrificed to collect the serum. VP28 and β-actin antisera were prepared in a similar manner, as described previously [55].

## Chitinase activity assay

The sub-cuticular tissue was collected at 48 h after WSSV infection and homogenized in PBS. After centrifugation at 12, 000 × *g* for 10 min, the supernatant was collected, and the protein concentration was determined as described above. Chitinase activity was determined using a Chitinase Assay Kit (Solarbio, Beijing, China; BC0825) by monitoring its ability to convert chitin to *N*-acetyl glucosamine. Generally, the reaction mixture (200 μL) consisted of 40 μL of sodium phosphate buffer (50 mM, pH 7.0), 80 μL of colloidal chitin (0.5%), and 80 μL of the test sample. The mixture was incubated at 37˚C for 1 h. After terminating the reaction by boiling for 5 min, the mixture was centrifuged at 12, 000 × *g* for 10 min. The resultant supernatant (100 μL) was mixed with 20 μL of 3,5-dinitrosalicylic acid and incubated for 5 min at 100˚C. The absorbance at 585 nm was monitored using a Multiskan FC microplate reader (Thermo Fisher Scientific, Waltham, MA, USA). A standard curve generated using gradient-diluted *N*-

acetyl glucosamine was used to determine the amount of reducing sugar. One unit (U) represents the chitinase activity necessary to convey the colloidal chitin to 1 μg of reducing sugar for 1 h at 37˚C. Chitinase activity is expressed as U per mg of sample protein.

## Application of inhibitor and chemicals

C75 (Selleck Chemicals, Houston, TX, USA; S9819) was dissolved in PBS containing 10% DMSO and 2% Tween-80. Each shrimp was injected with 5 μg of inhibitor. MA (Sigma-Aldrich; 70079), PA (Sigma-Aldrich; S4751), and SA (Sigma-Aldrich; P0500) were dissolved in DMSO. Each shrimp was injected with 5 μg of the fatty acid into the shrimp hemocoel. The corresponding solvent was used as the control.

## Analysis of expression profiles

Reverse transcription-PCR (RT-PCR) was used to study the distribution of chitinase transcripts using gene-specific primers (listed in S1 Table). The PCR procedure consisted of an initial incubation at 94˚C for 3 min, 30 cycles of 94˚C for 30 s, 54˚C for 30 s, and 72˚C for 30 s, and finally 72˚C for 10 min. The PCR products were analyzed using 1.5% agarose gel electrophoresis.

Quantitative real-time PCR (qPCR) was used to determine gene expression profiles after certain treatments using the primers shown in S1 Table and the RT-PCR-prepared cDNA as the template. The qPCR was performed using iQ SYBR Green Supermix (Bio-Rad, Hercules, CA, USA; 170–8882) and the CFX96 Real-Time System (Bio-Rad). The qPCR cycling procedure was as follows: 94˚C for 3 min, 40 cycles of 94˚C for 10 s and 60˚C for 1 min, and final dissociation from 65˚C to 95˚C. The gene encoding β-actin was used as the internal reference (S2 Table). The results were processed using the $2^{-\Delta\Delta Ct}$ method.

## RNA interference (RNAi)

Partial DNA fragments specific for certain genes and fused with the T7 promoter were amplified using the specific primers listed in S1 Table and were used as templates to synthesize dsRNA using a T7 RNAi Transcription Kit (Vazyme, Nanjing, China; TR102) according to the manufacturer's instructions. dsRNA specific for the green fluorescent protein (*GFP*) gene was synthesized simultaneously as the control. dsRNAs at a dose of 5 μg/g of shrimp were injected into shrimp hemocoel. The epidermis samples were collected to assess the knockdown efficiency using qPCR or western blotting 24 h later. Subsequent tests were performed at 24 h after dsRNA application.

## Monitoring shrimp motor ability

The movement trajectory and escape distance of a shrimp after prodding were used to monitor motor ability. Each shrimp was transferred into a new glass tank (20 × 29 × 21 cm) containing fresh seawater at 72 h after WSSV infection or 48 h after MA injection. Coordinate paper was placed behind the tank to facilitate measuring distances. After allowing the shrimp to acclimatize for 30 min, a glass rod was used to prod the shrimp lightly. The movement of shrimp was recorded using a camera for 5 s. The video was imported into ImageJ (NIH, Bethesda, MD, USA) and the Animal Tracker plugin was used to analyze the shrimp swimming trajectory. The escape distance of shrimp was calculated based on the swimming trajectory with the help of the coordinate paper. The tests were conducted in a quiet room to prevent distraction and unexpected behavior.

### Electrophoretic mobility shift assay (EMSA)

EMSA was performed to check the interaction of E75 with putative E75 binding sites in the promoter region of *Chi2*. Oligonucleotide probes were synthesized (Sangon Biotech, Shanghai, China), and labeled with biotin using an EMSA Probe Biotin Labeling Kit (Beyotime, Shanghai, China; GS008). EMSA was performed using a Chemiluminescent EMSA Kit (Beyotime; GS009), as described by the manufacturer. Generally, 2 μg of purified recombinant E75 was incubated with 5 ng of the probes at 25˚C for 20 min in a binding buffer. In the competition binding assays, excess unbiotinylated probes were added to the mixture. The samples were analyzed on 6% acrylamide gels in $0.5 \times$ Tris-borate-EDTA buffer and then electroblotted onto a nylon membrane. The labeled DNA was detected and visualized using a chemiluminescent biotin-labeled nucleic acid detection kit.

### Chromatin immunoprecipitation (ChIP) assay

A ChIP assay was performed to detect the transcriptional regulation of *Chi2* by E75 using the ChIP Assay Kit (Beyotime; P2078) according to the manufacturer's instructions. The sub-cuticle epidermis was collected at 48 h after WSSV infection or at 12 h after MA injection and was used as the pool for ChIP. The immunoprecipitates were analyzed using RT-PCR with primers (S1 Table) specific for the fragments containing E75 binding sites.

### Metabolomics analysis

The sub-cuticle epidermis was collected at 24 h post-WSSV infection, and placed into a sterile Eppendorf tube with 40 μL of internal standard (0.3 mg/mL L-2-chloro-phenylalanine) and 360 μL of pre-cooled methanol. The mixtures were ultrasonicated in ice water for 10 min and placed at -20˚C for 30 min. The extract was centrifuged at 4˚C and $13,000 \times g$ for 10 min to isolate the supernatant, which was further dried in a freeze concentration centrifugal dryer. The metabolomic analysis was then conducted by Shanghai Luming Biological Technology Co., Ltd. (Shanghai, China). Generally, for LC-MS analysis, an ACQUITY UPLC I-Class plus (Waters Corporation, Milford, MA, USA) fitted with a Q-Exactive mass spectrometer equipped with a heated electrospray ionization (ESI) source (Thermo Fisher Scientific) was used to analyze the metabolic profile in both ESI positive and ESI negative ion modes. For GC-MS analysis, an Agilent 7890B gas chromatography system coupled to an Agilent 5977A MSD system (Agilent Technologies Inc., Santa Clara, CA, USA) was used. After normalizing the results by data normalization, redundancy removal, and peak merging, a data matrix was obtained for subsequent analyses. Two-tailed Student's *t*-tests were used to evaluate differences in metabolites between two groups. Differential metabolites were identified when variable importance in projection (VIP) $> 1.0$, $p < 0.05$, and fold change $> 2$.

### Molecular docking

The structure of aa 175–426 of E75 was predicted using AlphaFold2 [56]. The predicted loop region (aa 220–239) with low confidence scores was manually removed. The obtained structure model was used as the receptor for molecular docking with fatty acids using AutoDock Vina 1.1.2 [57]. The structures of the protein-ligand complexes were visualized using PyMOL version 2.4.1 (Schrödinger, LLC, NY, USA).

### Isothermal titration calorimetry (ITC) assay

The ITC assay was performed to determine the interaction between MA and the E75 LBD using MICROCAL PEAQ-ITC (Malvern Panalytical, Malvern, UK). MA (800 μM) was

pumped in a syringe, and rE75 (10 μM) was placed into the sample cell. MA solution (2 μL) was injected into the sample cell and stirred at 750 rpm for 150 s, followed by 18 injections. rTag expressed by the empty expression vector was used as the control. Three independent repeats were performed.

## Statistical analysis

Images are representative of three independent repeats, and all bar charts show the mean ± SD derived from at least three independent experiments. At least five shrimp were used to prepare each sample. A two-tailed Student's *t*-test was used for comparisons using GraphPad Prism 8 (GraphPad Inc., La Jolla, CA, USA), and a significant difference was accepted at $p < 0.05$.

## Supporting information

**S1 Fig. Expression profiles and domain architecture of the kuruma shrimp chitinase family.** Left, tissue distribution and expression changes after WSV infection of shrimp chitinases, as analyzed using RT-PCR. The images are representative of three independent replicates. Middle, domain architecture of the chitinase family analyzed using the online tool SMART (smart.embl.de). Right, accession numbers for the chitinase sequences.
(TIF)

**S2 Fig. Temporal expression profiles of *Chi2*, *Chi13*, and *E75* after WSSV infection.** (A) WSSV VP28 level in the sub-cuticle epidermis. Shrimp were injected with a WSSV inoculum, and VP28 levels were analyzed using western blotting with β-actin as the internal reference. The blot images are representative of three independent replicates. (B) Temporal expression profiles of *Chi2*, *Chi13*, and *E75* in the epidermis after WSSV infection analyzed by qRT-PCR with β-actin as the internal reference. The fold change in expression levels in the WSSV group compared with the PBS group at each time point is shown. Mean ± SD of four independent replicates. Different characters indicate significant differences, analyzed by one-way ANOVA. (C) E75 level in the sub-cuticle epidermis. Shrimp were injected with a WSSV inoculum, and E75 levels were analyzed using western blotting with β-actin as the internal reference. Images of blots are representative of three independent replicates.
(TIF)

**S3 Fig. RNAi efficiencies of selected genes.** Indicated dsRNAs at a dose of 5 μg/g of shrimp were injected into shrimp hemocoels. The knockdown efficiency in the sub-cuticle epidermis was determined by using qPCR at 24 h after dsRNA application. Mean ± SD from four independent replicates, unpaired Student's *t*-test, ***$p < 0.001$. E75 levels were also analyzed using western blotting with β-actin as the internal reference. Images of blots are representative of three independent replicates.
(TIF)

**S4 Fig. Clear separation of epidermis metabolite profiles between the WSSV-infection group and control group, as shown by the PLS-DA score plot.**
(TIF)

**S5 Fig. Variation in shrimp hemolymph 20-hydroxyecdysone (20E) levels after WSSV infection.** 20E concentration was determined using a 20E ELISA kit (Mlbio, Shanghai, China; ML521987) according to the manufacturer's instructions. Absorbance at 450 nm was monitored using a Multiskan FC microplate reader (Thermo Fisher Scientific, Waltham, MA, USA). The amount of 20E was determined using a standard curve generated from a gradient

dilution of the standard samples.
(TIF)

**S1 Video. Videos of shrimp movement.** Shrimp was transferred into a new glass tank containing fresh seawater at 72 h after WSSV infection or 48 h after MA injection, and was allowed to acclimatize for 30 min. A glass rod was used to prod the shrimp lightly, and the video was recorded.
(PPTX)

**S1 Table. Primers and probes used in this study.**
(DOCX)

**S2 Table. Original values for all graphs.**
(XLSX)

## Author Contributions

**Conceptualization:** Xian-Wei Wang.

**Data curation:** Xin-Xin Wang.

**Formal analysis:** Xin-Xin Wang.

**Funding acquisition:** Xian-Wei Wang.

**Investigation:** Xin-Xin Wang, Ming-Jie Ding, Jie Gao, Ling Zhao.

**Methodology:** Xin-Xin Wang, Jie Gao, Ling Zhao, Xian-Wei Wang.

**Project administration:** Xian-Wei Wang.

**Resources:** Rong Cao, Xian-Wei Wang.

**Software:** Xin-Xin Wang, Xian-Wei Wang.

**Supervision:** Xian-Wei Wang.

**Validation:** Xin-Xin Wang, Jie Gao.

**Visualization:** Xin-Xin Wang, Jie Gao, Ling Zhao, Xian-Wei Wang.

**Writing – original draft:** Xin-Xin Wang, Xian-Wei Wang.

**Writing – review & editing:** Xian-Wei Wang.

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
