## [Decision Letter · Decision Letter 0]

19 Feb 2024

Dear Dr. Wang,

Thank you very much for submitting your manuscript "Modulation of host lipid metabolism by virus infection leads to damage to the exoskeleton in shrimp" for consideration at PLOS Pathogens. As with all papers reviewed by the journal, your manuscript was reviewed by members of the editorial board and by several independent reviewers. In light of the reviews (below this email), we would like to invite the resubmission of a significantly-revised version that takes into account the reviewers' comments.

We cannot make any decision about publication until we have seen the revised manuscript and your response to the reviewers' comments. Your revised manuscript is also likely to be sent to reviewers for further evaluation.

Sincerely,

Hai-peng Liu, Ph.D

Guest Editor

PLOS Pathogens

Patrick Hearing

Section Editor

PLOS Pathogens

Michael Malim

Editor-in-Chief

PLOS Pathogens

orcid.org/0000-0002-7699-2064

Reviewer's Responses to Questions

**Part I - Summary**

Reviewer #1: In this manuscript, the authors found that white spot syndrome virus (WSSV) infection can alter shrimp lipid metabolism mediated by E75-Chi2-Chitin signaling to damage to the exoskeleton. Some aspects or conclusions should be carefully verified or needs more experimental evidences.

Reviewer #2: In the current manuscript, the authors demonstrated a serial of detailed evidence to show the mechanisms of the loose and softening of exoskeleton in shrimp after WSSV infection and uncovered the relation of chitinase induced exoskeleton hydrolysis with the lipogenesis of LCFAs metabolism via an E75 receptor. Overall, the MS was written well, and the experimental design as well as the results presentation are all very good. It shed the valuable lights on better understanding of the pathogenicity of WSSV, and also provide useful directions for preventing the WSSV-caused disease in shrimp aquaculture industry.

Reviewer #3: The study by Wang et al. sheds light on the molecular mechanism underlying the damage to shrimp exoskeletons following viral infection. The authors demonstrate that WSSV infection reduces chitin levels in the cuticle by upregulating the expression of chitinase Chi2 in the epidermis. They further identify the nuclear receptor E75 as a key player in the WSSV-induced upregulation of Chi2. Moreover, WSSV infection increases host lipid profiles, with enriched saturated long-chain fatty acids (LCFAs) serving as a switch for E75's transcriptional activity. The authors also show that inhibiting LCFA generation improves the mechanical properties of shrimp exoskeletons. By elucidating this intriguing mechanism, the study significantly advances our understanding of WSSV pathogenesis and arthropod-virus interactions. Overall, the study is well-executed, and the results are compelling and robust. These findings hold relevance for a wide readership of PLoS Pathogens. However, certain concerns need to be addressed before publication in this journal.

**Part II – Major Issues: Key Experiments Required for Acceptance**

Reviewer #1: Major aspects:

1. One of the major defects is that all knockdown experiments did not detect protein levels, because that mRNA level knockdown does not represent protein level knockdown. It is necessary to confirm the knockdown efficiency of relevant protein levels, especially chi2 and E75.

2. Shrimp motor abilities are key parts of these results, so the authors should provide more intuitive and convincing evidences, for example, behavioral devices and videos or other experimental records. Besides, examine how the data is analyzed and interpreted, particularly the connection between lipid metabolism and WSSV infection in shrimp. The conclusions should be strongly supported by the data.

3. More molecular biological and in vivo experiments are needed to confirm the LCFAs-E75-Chi2-Chitin cascade in shrimp. More importantly, it doesn’t know how the LCFAs-E75-Chi2-Chitin is activated by WSSV infection or can also activated by other pathogens such as DIV1 or bacterial infection, so does this have uniqueness for WSSV or not?

4. Why choose the only 72 hpi for testing the mechanical performance of exoskeleton, impaired motor ability and others. How the expression profile of Chi2 and E75 in response to WSSV. Whether the dynamic process of exoskeleton is related to the expression of Chi2 and E75 induced by WSSV.

Reviewer #2: NA

Reviewer #3: 1. The authors proposed that the increase of LCFAs led to the enhanced Chi2 transcription by elevating the transcriptional activity of E75. However, E75 expression may also be upregulated by WSSV infection or LCFAs application, and the increase of E75 amount may also lead to the increase of Chi2 transcription. Though the authors have proved that there is no change in E75 mRNA level after WSSV infection (Fig. S2), a protein level data is needed to fully support the current hypothesis. Moreover, the specificity of E75 antibody should also be verified.

2. It seems that E75 regulates the expression of Chi2 in an ecdysone-independent manner. Is there any change in the ecdysone concentration after WSV infection? Pathogenic infection usually leads to the variation of ecdysone titer in insects. How about the case in shrimp? I suggest to determine the ecdysone titer to check whether WSSV infection would interfere with the ecdysone system.

3. This study revealed how shrimp exoskeleton is impaired by viral infection. What is the biological significance of this mechanism? Is there any possibility to apply the current finding into the disease control in shrimp culture? The authors can make a deeper discussion on this issue.

**Part III – Minor Issues: Editorial and Data Presentation Modifications**

Reviewer #1: 1. Ensure that the methods used for Chitinase activity assay, RNA interference, Monitoring shrimp motor ability, etc. are clearly described. Detailed methods allow for better reproducibility and understanding of the techniques involved. For Chitinase activity assay, how to set internal parameters to eliminate differences between groups?

2. In many places, the p value should be italic. The manuscript should be polished by an English-native speaker.

Reviewer #2: For the results (Line 190 -201), Fig. 2D, 2E, 2F, 2G and 2h, all should be Fig. 3D, 3E,3F, 3G, AND 3H.

Reviewer #3: 1. Though the authors have provided the primers for all chitinases, the accession number of these sequences should be provided.

2. How about the RNAi efficiency for the genes in Fig. 3B? The authors should validate the RNAi-mediated knockdown efficiency, and show the data in Fig. S3.

3. Is the chitin probe self-made or a commercial one? The source of this probe should be provided in detail.

4. Fig. 6G, why there is a shift for the green line?

5. Fig. S2A, there seems an error when preparing the figure. A symbol is below the panel of actin.

6. Some essential information is lacking in the figure legends. The reviewer has to jump back to the section of Materials and methods from time to time to understand at what conditions were the experiments performed. For example, the excitation source of the Raman spectra should be detailed in the figure legends.

7. This manuscript would benefit from a careful proof reading since a series of grammar errors and typos are present in the text.

PLOS authors have the option to publish the peer review history of their article (what does this mean?). If published, this will include your full peer review and any attached files.

Reviewer #1: No

Reviewer #2: **Yes: **JUN LI

Reviewer #3: No
---

## [Decision Letter · Decision Letter 1]

29 Apr 2024

Dear Dr. Wang,

We are pleased to inform you that your manuscript 'Modulation of Host Lipid Metabolism by Virus Infection Leads to Exoskeleton Damage in Shrimp' has been provisionally accepted for publication in PLOS Pathogens.

Best regards,

Hai-peng Liu, Ph.D

Guest Editor

PLOS Pathogens

Patrick Hearing

Section Editor

PLOS Pathogens

Michael Malim

Editor-in-Chief

PLOS Pathogens

orcid.org/0000-0002-7699-2064

Reviewer Comments (if any, and for reference):

Reviewer's Responses to Questions

**Part I - Summary**

Reviewer #1: The authors have well revised this manuscript.

Reviewer #2: The authors have responded all the questions and concerns from reviewers, I am satisfied to the responses and the revised version of the manuscript. It is acceptable for considering to be published in the journal of Plos Pathogen.

Reviewer #3: The authors have addressed my concerns and I have no further questions.

**Part II – Major Issues: Key Experiments Required for Acceptance**

Reviewer #1: (No Response)

Reviewer #2: The authors addressed all the concerns for the reviewers by providing extra evidence to support their claimed idea.

Reviewer #3: NA

**Part III – Minor Issues: Editorial and Data Presentation Modifications**

Reviewer #1: (No Response)

Reviewer #2: Satisfied for the revision.

Reviewer #3: NA

PLOS authors have the option to publish the peer review history of their article (what does this mean?). If published, this will include your full peer review and any attached files.

Reviewer #1: No

Reviewer #2: No

Reviewer #3: No

---

## [Editor Report · Acceptance letter]

6 May 2024

Dear Dr. Wang,

We are delighted to inform you that your manuscript, "Modulation of Host Lipid Metabolism by Virus Infection Leads to Exoskeleton Damage in Shrimp," has been formally accepted for publication in PLOS Pathogens.

Best regards,

Michael Malim

Editor-in-Chief

PLOS Pathogens

orcid.org/0000-0002-7699-2064